# MAGNET: AUGMENTING GENERATIVE DECODERS WITH REPRESENTATION LEARNING AND INFILLING CAPABILITIES

## ABSTRACT

While originally designed for unidirectional generative modeling, decoder-only large language models (LLMs) are increasingly being adapted for bidirectional modeling. However, these unidirectional and bidirectional models are typically trained independently with distinct objectives (generation or representation learning) thereby missing the potential opportunity for one objective to enhance the other. In this work, we introduce MAGNET, an adaptation of decoder-only LLMs that enhances their capabilities in generating robust representations and infilling missing text spans, while maintaining coherent and non-repetitive text generation. MAGNET employs three self-supervised training objectives and introduces an attention mechanism that combines bidirectional and causal attention, enabling unified training across all objectives. We show that LLMs adapted using MAGNET can outperform powerful text encoders on token-level and sentence-level representation learning tasks. We also demonstrate that MAGNET enhances the base LLM's ability to generate contextually appropriate text infillings by enabling it to take future context into consideration. Lastly, we show that, unlike other bidirectional language models for representation learning, the LLMs adapted using MAGNET can still perform open-ended text generation.

## 1 INTRODUCTION

Language models are computational models designed to understand and generate human language. They have transformed natural language processing, powering applications such as text annotation, machine translation, summarization, speech recognition, and dialogue systems. Traditionally, language models are categorized into three main types: (1) *Encoder-only models* (Devlin et al., 2019; Liu et al., 2019; Lan et al., 2019; Sanh et al., 2019), which focus on encoding the input into fixed-dimensional representations and excel in tasks like sentiment analysis (sentence-level classification) and named entity recognition (token-level classification). (2) *Decoder-only models* (Brown et al., 2020; Hoffmann et al., 2022; Touvron et al., 2023a;b; Scao et al., 2022; Penedo et al., 2023), which are adept at generating coherent text, thereby specializing in tasks like creative content generation and dialogue systems. (3) *Encoder-decoder models* (Raffel et al., 2019; Lewis et al., 2019), wherein an encoder understands the input and a decoder generates the corresponding output, making this architecture suitable for tasks like machine translation and summarization.

Recently, the NLP community has increasingly embraced decoder-only architecture (Large Language Models or LLMs) due to their efficient training and scalability to larger datasets, resulting in enhanced performance across various tasks. However, since these models are trained using causal attention and lack bidirectional context, they are less suitable for tasks like (1) sentiment analysis and named entity recognition, which require understanding contextual representations of sentences or words, and (2) text infilling, where predicting missing text spans must maintain coherence with subsequent context. With the goal of leveraging the scalable decoder-only LLMs, some recent efforts (BehnamGhader et al., 2024; Li & Li, 2023; Li et al., 2023; Duki'c & vSnajder, 2024; Du et al., 2021; Donahue et al., 2020) have aimed to adapt decoder-only LLMs for these tasks. However, as illustrated in Figure 1, methods that enhance LLMs for text infilling do not enable them to function as effective text encoders, while approaches focused on representation learning hinder their generative capabilities.

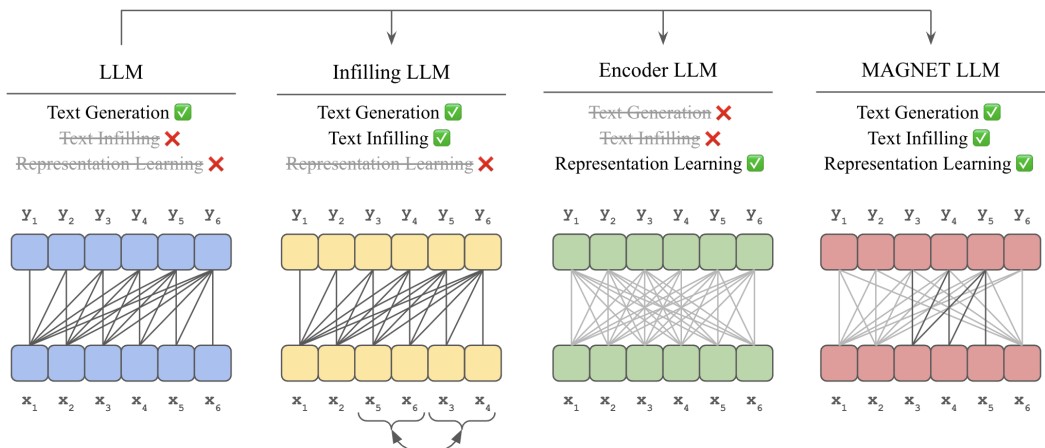

Figure 1: Traditionally, LLMs are trained for text generation using unidirectional attention between the input $x$ and output $y$ (depicted by black lines), whereas text encoders are trained for representation learning using bidirectional attention (depicted by gray lines). MAGNET adapts the attention mechanism of LLMs to combine both unidirectional and bidirectional attention, enhancing them with representation learning and infilling capabilities, while retaining their core generative functions.

In this work, we propose MAGNET (Modified Attention for Generation and Encoding of Text), an adaptation of decoder-only LLMs for (1) generating robust sentence-level and token-level representations, (2) infilling missing spans of text while preserving coherence with bidirectional context, and (3) performing open-ended text generation without excessive repetition of words or phrases. To achieve this, we use three self-supervised training objectives: (1) a *masked modeling objective* to learn token-level representations, (2) a *contrastive objective* to learn sentence-level representations, and (3) a *missing-span generation objective* to infill text and retain generative capabilities. To facilitate simultaneous training across all these objectives, we deploy a specially crafted attention mask that is a combination of bidirectional and causal attention.

Without any model-specific design or loss of generality, we apply MAGNET to LLaMA-2-7B (Touvron et al., 2023b). We demonstrate that the proposed method requires simple modification and fine-tuning of an off-the-shelf LLM to augment it with representation learning and infilling capabilities. Our results show that MAGNET-adapted LLaMA-2-7B outperforms other methods that adapt the same model for token-level and sentence-level representation learning tasks[1]. Further, we also show that MAGNET significantly improves the infilling capability of the LLM by enabling it to consider bidirectional context. Lastly, we analyze the repetition problem in text generated by text encoders and demonstrate that MAGNET-adapted models are significantly better at open-ended text generation than other text encoders.

## 2 RELATED WORKS

**Representation Learning.**

Text representation learning focuses on understanding contextual relationships within sentences. Traditionally, encoder models dominated this field due to their bidirectional context modeling, using masked language modeling for token-level representations (Devlin et al., 2019; Liu et al., 2019; He et al., 2020; Clark et al., 2020; He et al., 2021) and special tokens with similarity-based optimization for sentence-level understanding (Gunel et al., 2020; Reimers & Gurevych, 2019; Wu et al., 2020; Carlsson et al., 2021; Gao et al., 2021; Wei et al., 2020). Recent work has explored adapting decoder-only LLMs for representation learning, either by using last-token or mean-pooled representations (Neelakantan et al., 2022; Wang et al., 2023), or by fine-tuning with masked modeling (BehnamGhader et al., 2024) or label supervision (Li et al., 2023; Duki'c & vSnajder, 2024). While some approaches modify the decoder's causal attention to be bidirectional (BehnamGhader et al.,

---

[1]It is to be noted that these other methods exclusively adapt the model for learning representations, while MAGNET trains for other objectives as well.

2024; Muennighoff et al., 2024; Li & Li, 2023; Duki'c & vSnajder, 2024), this often compromises the model's text generation abilities. In contrast, MAGNET employs a hybrid attention mechanism that combines causal and bidirectional attention, enabling both robust representation learning and preserved generation capabilities.

**Text Infilling.** Text infilling requires considering both left and right context when generating text in the middle of a sequence. Encoder-decoder models (Raffel et al., 2019; Lewis et al., 2019; Kalinsky et al., 2023) can handle this task by encoding available context and decoding infilled text. Other approaches have extended masked language modeling to perform span infilling (Joshi et al., 2019; Shen et al., 2023; 2020). Decoder-only models have also been adapted for infilling through various strategies: training models to directly fill marked blanks (Donahue et al., 2020; Du et al., 2021), rearranging training examples to align with infilling objectives (Bavarian et al., 2022; Yang et al., 2019; Aghajanyan et al., 2022; Fried et al., 2022), or using dual generation from both ends of a sentence until convergence (Nguyen et al., 2023; Serdyuk et al., 2017). However, while these approaches successfully enhance LLMs with infilling capabilities, none have attempted to simultaneously equip them with both infilling and representation learning abilities, as done by MAGNET.

**Unifying Text Understanding and Generation.** Prior works on unifying natural language understanding and generation within a single framework usually focus on proposing pretraining objectives and task formulations. These approaches typically extend traditional masked language modeling, with innovations like permutation-based objectives for bidirectional context modeling (Yang et al., 2019), autoregressive blank infilling (Du et al., 2021), multi-directional attention masks (Dong et al., 2019), and sequence-to-sequence pretraining (Song et al., 2019; Raffel et al., 2019). However, these approaches require pretraining new networks from scratch, despite decoder-only models demonstrating exceptional scalability and effectiveness. Instead of starting from scratch, we propose a parameter-efficient method that builds upon the rich representations already learned by existing large language models, transforming them into a unified framework for representation learning, text infilling, and text generation.

## 3 METHOD

Decoder-only models, based on the Transformer architecture (Vaswani et al., 2017), process input sequences through successive blocks of multi-head self-attention, feed-forward networks, and layer normalization. The self-attention mechanism converts the input $\mathbf{x} \in \mathbb{R}^{l \times d}$ into queries $\mathbf{Q}$, keys $\mathbf{K}$, and values $\mathbf{V}$ using a linear projections, and computes attention using the formula:

$$\texttt{Attn}_i(\mathbf{Q}, \mathbf{K}, \mathbf{V}) = \texttt{softmax}\left(\frac{\mathbf{Q}\mathbf{K}^{\mathrm{T}} + \mathbf{M}}{\sqrt{d_k}}\right)\mathbf{V} \tag{1}$$

where $\texttt{Attn}_i$ is the $i^{th}$ head of the multi-head self-attention, $d_k$ represents the dimensionality of the keys/queries, and $\mathbf{M}$ represents the causal mask. This causal mask $\mathbf{M}$ for an autoregressive LLM is an $l \times l$ strictly upper triangular matrix with the upper triangle set to $-\infty$, as shown in Figure 2a. $\mathbf{M}$ ensures that the softmax operation assigns an attention weight of zero to the future positions in the sequence, which in turn ensures that each token $i$ can only attend to itself and tokens that precede it in the sequence.

MAGNET seeks to update the causal attention mechanism of an LLM by incorporating elements of bidirectionality and thereafter fine-tunes the model using unsupervised objectives. We look at the modifications to the attention mechanism in Section 3.1 and the training objectives in Section 3.2.

### 3.1 MODIFYING ATTENTION

MAGNET updates the causal attention mechanism of an LLM to introduce bidirectional capabilities within segments of the input sequence, as illustrated in Figure 2. After the input text is tokenized for the language model, we categorize each token as either *context tokens* or *span tokens*:

**Context tokens.** Each context token (shown in blue in Figure 2) attends to all other context tokens within the sequence. In our implementation, the attention mask is designed with 0s at output positions

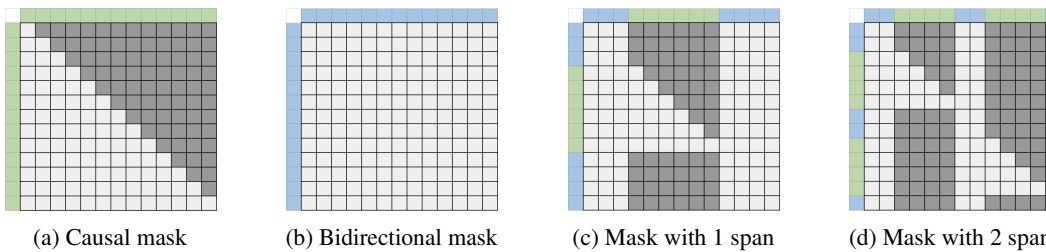

(a) Causal mask      (b) Bidirectional mask      (c) Mask with 1 span      (d) Mask with 2 spans

Figure 2: Matrices illustrating attention masks for different types of attention mechanisms. The rows of the matrices correspond to the query tokens and the columns correspond to the key tokens. Light gray cells indicate 0, dark gray cells represent $-\infty$, green marks span token positions, and blue marks context token positions. Each context token attends to every other context token, and each span token attends to all context tokens and the preceding span tokens in the same span.

corresponding to context tokens, allowing each context token to access information from every other context token. This transformation shifts the original unidirectional LLM into a bidirectional model.

**Span tokens.** The span tokens (shown in green in Figure 2) represent a contiguous span of input tokens that attend to all context tokens and have causal attention among themselves. By enabling span tokens to access surrounding context, we effectively convert the original LLM into an infilling language model. Additionally, the causal attention among span tokens preserves the LLM's generative capabilities, which could be compromised if bidirectionality is fully unlocked (see Section 4.4 for further details).

During training, an input sequence includes one or more spans of span tokens surrounded by context tokens. During inference, the attention mechanism can operate in three modes: (1) fully causal/unidirectional for open-ended text generation tasks, (2) fully bidirectional for tasks requiring representation learning, or (3) a combination of causal and bidirectional for text infilling.

## 3.2 TRAINING OBJECTIVES

MAGNET fine-tunes an off-the-shelf LLM using three self-supervised objectives aimed at enhancing the model's ability to learn contextually-rich representations and autoregressively fill in missing spans of text. These objectives are illustrated in Figure 3 and discussed below.

### 3.2.1 MASKED NEXT TOKEN PREDICTION (MNTP)

MNTP enables the model to realize its newly enabled bidirectional attention capability. The task is defined as follows: Given an input sequence $\mathbf{x} = (x_1, x_2, ..., x_L)$, we select a fraction of the input tokens for masking and train the model to predict these masked tokens. In our setup, we find that selecting 20% of the input tokens for masking works well. Further, following Devlin et al. (2019), we replace 80% of the selected tokens with a `[MASK]` token, 10% with a random token from the model's vocabulary, and leave the remaining 10% unchanged. Since LLMs are trained to predict the next token in a sequence, we use the token representations from position $l$ to predict a masked token at position $l + 1$[2]. MNTP is optimized using categorical cross-entropy loss:

$$\mathcal{L}_{\text{MNTP}} = -\frac{1}{NL} \sum_{n=1}^{N} \sum_{l=1}^{L} \sum_{v=1}^{V} \mathbb{1}_{\text{mask}}(l+1) \cdot (y_{lv}^{(n)} \log(\hat{y}_{lv}^{(n)})) \tag{2}$$

where, $N$ denotes batch size, $L$ denotes the sequence length, $V$ denotes vocabulary size, $\mathbb{1}_{\text{mask}}(l+1)$ is 1 if position $l + 1$ is masked and 0 otherwise, and $y_{lv}$ and $\hat{y}_{lv}$ represent the true and predicted probabilities for the $v^{th}$ token in the vocabulary at position $l$ in the sequence. It is to be noted that this task is conducted exclusively with the context tokens.

---

[2]In Appendix E, we explore the possibility of using masked token prediction (MTP) objective, where the output at token $l$ predicts the masked token at position $l$.

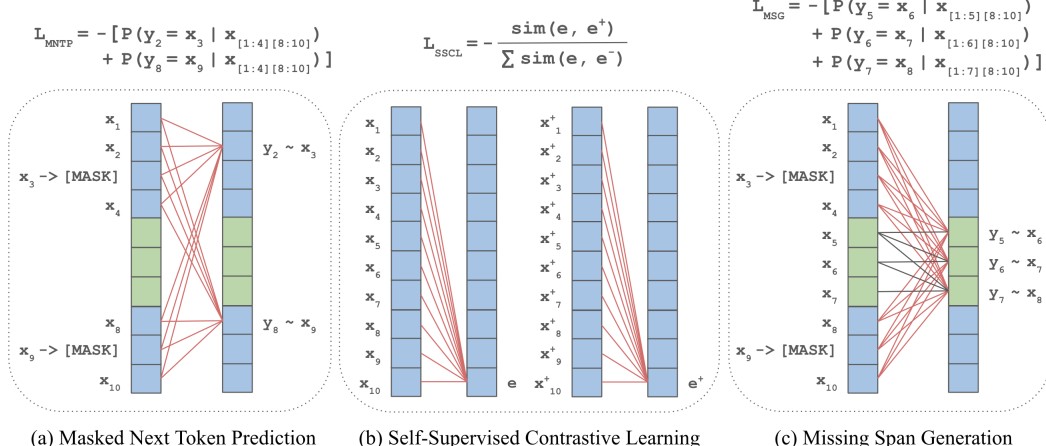

(a) Masked Next Token Prediction     (b) Self-Supervised Contrastive Learning     (c) Missing Span Generation

Figure 3: MAGNET training objectives include: (a) Masked next token prediction, which is applied on the output corresponding to the token preceding the masked context token. (b) Self-supervised contrastive learning, which is applied on the model's representation corresponding to the last token. (c) Missing span generation, which is applied on the output corresponding to the span tokens. In this illustration, the red lines denote bidirectional attention and the black lines denote causal attention. Further, for (a) and (c), the output token $y_i$ is trained to predict the input token $x_{i+1}$, as denoted by "$y_i \sim x_{i+1}$"

### 3.2.2 SELF-SUPERVISED CONTRASTIVE LEARNING (SSCL)

Since LLMs are not explicitly trained to capture the entire input context and generate sentence-level representations, we employ SSCL to transform them into text encoders. The training objective is defined as follows: Given an input sequence $\mathbf{x}$, generate its augmented view $\mathbf{x}^+$ and align their encoded representations $\mathbf{e} = f(\mathbf{x})$ and $\mathbf{e}^+ = f(\mathbf{x}^+)$ in the embedding space, while distancing them from the encodings $\mathbf{e}^- = f(\mathbf{x}^-)$ of other input sequences $\mathbf{x}^-$ in a training batch. Specifically, we employ paraphrasing (Damodaran, 2021) to generate augmented views of an input, and add an instruction "*Given the sentence, find its representation:*" to the training examples (Jiang et al., 2023). Then, we use the representations corresponding to the last token ([EOS]) of the final hidden states as the sentence encoding. Our choice of using the last token representation as the encoding is guided by the fact that MAGNET optimizes simultaneously for token-level and sentence-level representations. Since the last token's representation is not used for token-level optimization (because the representation of input token $i$ is given by output token $i - 1$), this choice enables us to disentangle the two representation learning tasks during joint training. We use InfoNCE (van den Oord et al., 2018) with in-batch negatives as the loss function:

$$\mathcal{L}_{\text{SSCL}} = -\frac{1}{N} \sum_{i=1}^{N} \log \frac{\exp(\mathbf{e}_i \cdot \mathbf{e}_i^+ / \tau)}{\sum_{j=1}^{N} \exp(\mathbf{e}_i \cdot \mathbf{e}_j^- / \tau)} \tag{3}$$

where, $N$ represents the batch size and $\tau$ denotes the temperature for logit scaling.

### 3.2.3 MISSING SPAN GENERATION (MSG)

MSG provides text infilling capabilities to the left-to-right autoregressive model. The task is defined as: Given a position $p$ and an input sequence $\mathbf{x} = (x_1, ..., x_p, x_q, ..., x_L)$, generate a plausible sequence of $m$ tokens $\mathbf{y} = (y_1, y_2, ..., y_m)$ that fits between $x_p$ and $x_q$. More specifically, in our training setup, this task entails predicting a span token $y_l$ conditioned on all context tokens in $\mathbf{x}$ and the preceding span tokens $x_{[1..l-1]}$. We train using categorical cross-entropy loss computed over the predicted span tokens:

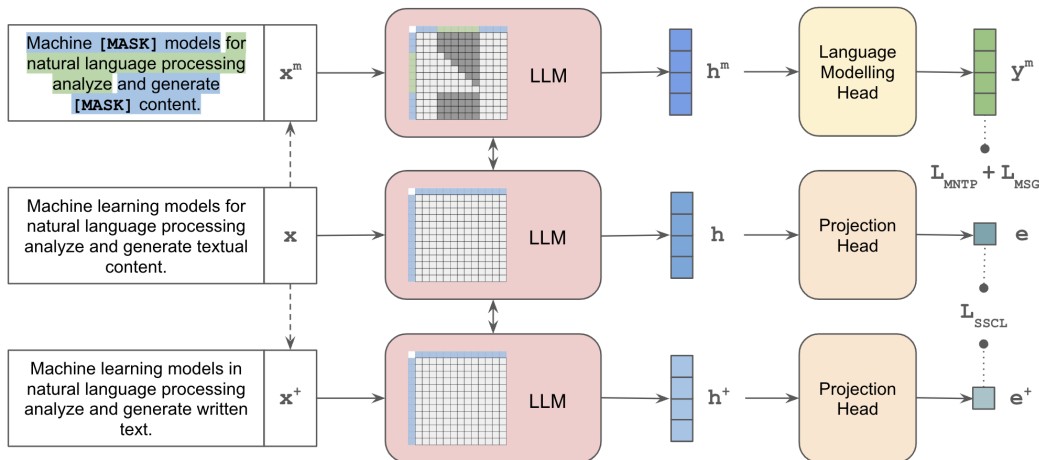

Figure 4: MAGNET processes three views of the input using different attention mechanisms within the same LLM. The model is trained (or fine-tuned) using three self-supervised learning objectives simultaneously to augment it with the ability to generate token-level and sentence-level representations, and perform text infilling tasks, while maintaining its original left-to-right text generation capability.

$$\mathcal{L}_{\text{MSG}} = -\frac{1}{N} \sum_{n=1}^{N} \sum_{l=1}^{L} \sum_{v=1}^{V} \mathbb{1}_{\text{span}}(l) \cdot (y_{lv}^{(n)} \log(\hat{y}_{lv}^{(n)})) \tag{4}$$

where, $N$ denotes batch size, $L$ denotes sequence length, $V$ denotes vocabulary size, $\mathbb{1}_{\text{span}}(l)$ is 1 if the token at position $l$ is a span token and 0 otherwise, and $y_{lv}$ and $\hat{y}_{lv}$ are the true and predicted probabilities for token $v$ in the vocabulary at position $l$ in the sequence. The standard next token prediction task of LLMs can be considered as a special case of this objective, wherein all input tokens are span tokens (and the attention mechanism reduces to causal attention). Thus, a beneficial side-effect of this task is that the model retains its text generation capability while learning representations.

## 3.3 APPROACH OVERVIEW

Figure 4 provides an overview of MAGNET. Starting with a training example $\mathbf{x}$, the process unfolds in two parallel streams – (1) One or more contiguous spans of $M$ tokens in $\mathbf{x}$ are marked as span tokens, while a fraction of the remaining tokens (context tokens) is masked to form $\mathbf{x}^{\text{m}}$. (2) $\mathbf{x}$ is augmented to get $\mathbf{x}^+$. The input sequences $\mathbf{x}$, $\mathbf{x}^{\text{m}}$ and $\mathbf{x}^+$ are processed by the base decoder model to produce hidden states $\mathbf{h}$, $\mathbf{h}^{\text{m}}$ and $\mathbf{h}^+$. From $\mathbf{h}^{\text{m}}$, a language modeling head generates $\mathbf{y}^{\text{m}}$, which is used to compute $\mathcal{L}_{\text{MNTP}}$ and $\mathcal{L}_{\text{MSG}}$. Parallelly, $\mathbf{h}$ and $\mathbf{h}^+$ are processed using a projection head to get $\mathbf{e}$ and $\mathbf{e}^+$, which are used to compute $\mathcal{L}_{\text{SSCL}}$. The overall loss function is given as:

$$\mathcal{L} = \lambda_1 \mathcal{L}_{\text{MNTP}} + \lambda_2 \mathcal{L}_{\text{SSCL}} + \lambda_3 \mathcal{L}_{\text{MSG}} \tag{5}$$

For processing $\mathbf{x}$ to get $\mathbf{x}^+$, the decoder utilizes a bidirectional attention mask, illustrated in Figure 2b. While processing $\mathbf{x}^{\text{m}}$, the decoder employs an attention mask similar to those depicted in Figures 2c and 2d. In some cases, when all input tokens are marked as span tokens, the attention mask reduces to causal attention, as shown in Figure 2a.

## 4 EXPERIMENTS

In this section, we show that applying MAGNET augments a decoder-only LLM with representation learning and infilling capabilities. All training details are mentioned in Appendix A. Additionally, we present ablation experiments demonstrating the benefits of training a bidirectional model with a causal objective in Appendix C.

Table 1: Results on word-level tasks. LLM2Vec (BehnamGhader et al., 2024) adapts the model using MNTP and SimCSE. LLM2Vec[MNTP] is an intermediate state of LLM2Vec that is trained only on MNTP. All numbers except those for MAGNET are taken from BehnamGhader et al. (2024).

| Model | Chunking | NER | POS-Tagging |
|---|---|---|---|
| *Encoder models* | | | |
| BERT-Large | 71.77 | 90.09 | 75.12 |
| XLNet-Large | 79.70 | 93.67 | 83.02 |
| DeBERTa-Large | 85.74 | 94.97 | 86.49 |
| StructBERT-Large | 89.99 | 97.31 | 90.86 |
| *Llama 2 models* | | | |
| LLaMA-2-7B | 88.23 | 96.59 | 91.53 |
| LLM2Vec | 89.66 | 96.05 | 90.53 |
| LLM2Vec[MNTP] | 91.61 | 97.16 | 92.61 |
| MAGNET | **92.64** | **98.31** | **93.34** |

Table 2: Results on STS tasks. The encoder models are trained using SimCSE and their results are taken from Gao et al. (2021). The results for LLaMa-2-7B are obtained using the last token embedding from the final hidden state as the sentence representation. The results for LLM2Vec and Echo Embeddings are taken from BehnamGhader et al. (2024) and Springer et al. (2024), respectively.

| Model | STS12 | STS13 | STS14 | STS15 | STS16 | STS-B | SICK-R | Avg |
|---|---|---|---|---|---|---|---|---|
| *Encoder models (finetuned using SimCSE)* | | | | | | | | |
| BERT-Base | 68.40 | 82.41 | 74.38 | 80.91 | 78.56 | 76.85 | 72.23 | 76.25 |
| RoBERTa-Base | 70.16 | 81.77 | 73.24 | 81.36 | 80.65 | 80.22 | 68.56 | 76.57 |
| RoBERTa-Large | **72.86** | 83.99 | 75.62 | **84.77** | **81.80** | 81.98 | 71.26 | 78.90 |
| *Llama 2 models* | | | | | | | | |
| LLaMA-2-7B | 50.98 | 74.02 | 62.86 | 67.09 | 71.03 | 63.56 | 67.22 | 65.25 |
| Echo Embeddings | 52.40 | 72.40 | 61.24 | 72.67 | 73.51 | 65.73 | 64.39 | 66.05 |
| LLM2Vec | 65.39 | 79.26 | 72.98 | 82.72 | 81.02 | 78.32 | 71.77 | 75.92 |
| MAGNET | 67.98 | **84.66** | **77.67** | 84.17 | 79.44 | **82.88** | **78.77** | **79.36** |

## 4.1 WORD-LEVEL TASKS

We evaluate the token-level representations on three tasks – (1) chunking, (2) named entity recognition, and (3) part-of-speech tagging – using the CoNLL-2003 dataset (Sang & Meulder, 2003). After applying the training objectives proposed in Section 3.2, we train a linear classifier on top of the frozen representations obtained from the last hidden state of the model. The word-level embeddings are obtained by averaging the representations of the tokens that make up that word. Further, the representation of the token at position $i$ is given by the embedding at position $i - 1$. More specifically, if $w_1 w_2 w_3$ denotes the original input sentence, $(x_{w_1 1}, x_{w_1 2}, x_{w_2 1}, x_{w_2 2}, x_{w_2 3}, x_{w_3 1})$ denotes its tokenized form, and $(y_{w_1 1}, y_{w_1 2}, y_{w_2 1}, y_{w_2 2}, y_{w_2 3}, y_{w_3 1})$ denotes the model's output, the representation for the word $w_2$ is computed as $(y_{w_1 2} + y_{w_2 1} + y_{w_2 2})/3$.

Table 1 compares MAGNET with powerful encoder models and LLM2Vec (BehnamGhader et al., 2024), a recent method for adapting decoder-only LLMs for representation learning. The second-best approach, LLM2Vec[MNTP], relies solely on MNTP for model adaptation. In contrast, MAGNET integrates both representation learning objectives (MNTP and SSCL) and generative objectives (MSG). The superior performance of MAGNET over LLM2Vec[MNTP], despite using the same training data, model, and parameters, highlights the synergistic advantages of a unified training strategy for word-level representation learning.

## 4.2 SENTENCE-LEVEL TASKS

We evaluate sentence-level representations on multiple semantic similarity and clustering benchmarks (Muennighoff et al., 2022). We perform these tasks using the representation corresponding to the

Table 3: Results on clustering tasks. The results for LLM2Vec and Echo Embeddings are taken from BehnamGhader et al. (2024) and Springer et al. (2024), respectively.

| Dataset | BiorxivClustering | TwentyNewsgroups | MedrxivClustering |
|---|---|---|---|
| Echo Embeddings | 25.92 | 23.42 | 24.30 |
| LLM2Vec | 34.69 | 30.76 | 29.49 |
| MAGNET | **35.10** | **53.31** | **30.21** |

Table 4: Results on the infilling tasks. We measure the perplexity (PPL) for sentence infilling and block-of-text infilling on ROC-Stories and Wikitext-103, respectively.

| Method | ROC Stories | Wikitext-103 |
|---|---|---|
| LLaMA-2-7B | 13.9347 | 22.0399 |
| MAGNET | **9.5161** | **15.4573** |

Table 5: Human evaluation results on the infilling tasks. The score denotes the percentage of infillings that were considered contextually appropriate by human evaluators.

| Method | Score |
|---|---|
| Unidirectional LLaMA-2-7B | 53.50 |
| Zero-Shot Setup | 5.50 |
| Five-Shot Setup | 54.50 |
| MAGNET | **62.00** |

last token ([EOS]), without performing any task-specific training. Further, task-specific instructions (Table 8) are used for extracting relevant representations (Su et al., 2022; Wang et al., 2023).

We compare the text encoding capabilities of MAGNET with other recently proposed methods for transforming decoder models into text encoders, viz. LLM2Vec (BehnamGhader et al., 2024) and Echo Embeddings (Springer et al., 2024). Table 2 shows the results on Semantic Textual Similarity (STS) task and Table 3 shows the results on clustering tasks. As can be seen, MAGNET outperforms other adaptation methods on STS and clustering tasks. As previously noted, the fact that MAGNET surpasses LLM2Vec suggests the potential benefit of a unified training approach. Additionally, MAGNET not only achieves better performance on text encoding tasks but also provides decoder models with more capabilities (e.g., infilling), which its competitors lack.

### 4.3 INFILLING TASK

To test infilling capabilities, we evaluate the perplexity (PPL) of LLaMA-2-7B and MAGNET-adapted LLaMA-2-7B on the test set of ROC Stories (Mostafazadeh et al., 2016) and Wikitext-103 (Merity et al., 2016). For ROC Stories, we randomly mask out a sentence from each 5-sentence story, while for Wikitext-103, we mask up to three spans with lengths ranging from 8 to 32 tokens. Following Donahue et al. (2020), we compute PPL only for the tokens comprising the original masked out spans. The results are presented in Table 4, and they show that the base model (LLaMA-2-7B) exhibits significantly higher perplexity for the masked spans compared to MAGNET, demonstrating that MAGNET effectively augments the base model with text infilling capabilities. This improvement is attributed to MAGNET's ability to incorporate all the surrounding contextual information when infilling text, which increases the likelihood of correctly predicting the original masked content.

We also conducted experiments using zero-shot and few-shot learning to enable LLaMA-2-7B to incorporate all the surrounding context when infilling a missing span. We explored various prompting strategies and found that while a zero-shot setup did not yield sensible infillings, a five-shot setup with descriptive prompts resulted in more context-aware infillings (refer Appendix B for details). For a comprehensive analysis, we conducted a human evaluation to compare the quality of infillings generated by the base model, its zero-shot variant, its few-shot variant, and its MAGNET adaptation. In this evaluation, we randomly sampled 100 stories from the ROC Stories dataset, masked out one of their middle sentences, and tasked the models with infilling the missing sentence. Two human annotators on Amazon Mechanical Turk then independently assessed whether each generated sentence was contextually appropriate and contributed to a coherent story. The results are presented in Table 5, and they show that the infillings generated by MAGNET-adapted model are significantly more coherent than those generated by the variants of the base model. We show some qualitative examples in Table 10, demonstrating that MAGNET successfully delivers relevant and contextually appropriate infillings.

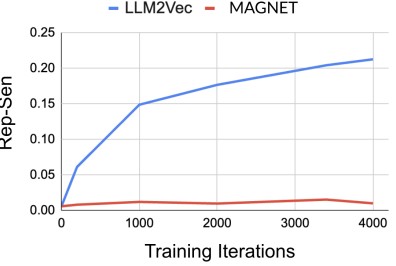

Figure 5: Analyzing the repetition problem in text generation. Both LLM2Vec and MAGNET are applied for 3400 iterations.

| Method | Wikitext-103 | | ROC Stories | |
|---|---|---|---|---|
| | **Rep-Sen** | **Rep-4** | **Rep-Sen** | **Rep-4** |
| LLaMA-2-7B | 0.0056 | 0.0601 | 0.0381 | 0.0163 |
| LLM2Vec | 0.2044 | 0.4747 | 0.2945 | 0.5243 |
| MAGNET | 0.0151 | 0.2047 | 0.0737 | 0.2573 |

Figure 6: LLM2Vec adaptation increases text repetition with more training, while no such trend is observed for MAGNET.

### 4.4 OPEN-ENDED TEXT GENERATION AND REPETITION PROBLEM

The repetition problem in text generation refers to the issue when generative models repeatedly produce the same phrases or sentences. Prior studies have identified that this issue often results from biases in the training data, limitations in the model's design, or standard likelihood training and inference (Holtzman et al., 2019; Welleck et al., 2019; Fu et al., 2020; Xu et al., 2022). In our study, we find that when generative decoder models are adapted into text encoders by enabling bidirectional attention (BehnamGhader et al., 2024; Li & Li, 2023; Li et al., 2023), the issue of repetition is significantly worsened. For example, Table 11 shows the texts generated (using greedy decoding) by the original LLaMA-2-7B and its adapted version for text encoding via LLM2Vec (BehnamGhader et al., 2024). We observe noticeable repetitions in the text generated by LLM2Vec-adapted-Llama, although the fine-tuning data (Wikitext-103) had almost no sentence-level repetitions (0.02%).

For quantitative detection of repetitions in texts generated by an LLM and its bidirectional adaptations, we compute *Rep-Sen* and *Rep-4* (as done by prior works analyzing the repetition problem (Holtzman et al., 2019; Welleck et al., 2019; Xu et al., 2022)):

$$Rep\text{-}Sen = 1.0 - \frac{|\text{ unique sentences }|}{|\text{ sentences }|} \qquad Rep\text{-}n = 1.0 - \frac{|\text{ unique n-grams }|}{|\text{ n-grams }|} \qquad (6)$$

Specifically, we create a *prefix-dataset* from the test sets of Wikitext-103 and ROC Stories, consisting of 5-word and single-sentence prefixes, respectively. The model is then tasked with autoregressively generating text based on these prefixes. Table 5 shows the repetition metrics for LLaMA-2-7B and its adaptations using LLM2Vec and MAGNET. As can be seen, MAGNET makes the base model significantly less prone to repeating sentences. For instance, for Wikitext-103, LLM2Vec makes LLaMA-2-7B 36.5 times more likely to repeat sentences, while MAGNET only makes it 2.7 times more likely. Further, as shown in Figure 6, the repetition problem exacerbates with additional iterations of LLM2Vec training, whereas no similar trend is observed with MAGNET.

We conjecture that LLM2Vec is significantly more prone to generating repetitive text because it exclusively focuses on learning representations with bidirectional attention. This training approach perhaps makes the decoder model somewhat similar to bidirectional LMs like BERT, which are known to repeat words when used for text generation (Table 11). MAGNET solves this issue by having autoregressive text generation as one of the training objectives.

## 5 CONCLUSION

In this work, we presented MAGNET, a method to transform causal LLMs into text encoders and infilling language models with bidirectional context-capturing ability. Through extensive experiments, we show that MAGNET uniquely equips LLMs with abilities that are beyond the scope of traditional text encoders or decoders. Thus, MAGNET shows the potential to unify text generation and text encoding within a single framework. Future research could explore scaling MAGNET to multimodal settings.

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

## A  TRAINING DETAILS

MAGNET fine-tunes LLaMA-2-7B using LoRA (Hu et al., 2021) with $r = 16$ and $\alpha = 32$. We use the AdamW optimizer with $\beta_1 = 0.9$, $\beta_2 = 0.999$ and $\epsilon = 1e - 8$, apply bfloat16 quantization, and use scaled-dot-product attention (SDPA). All experiments are performed on a single NVIDIA A100 GPU, with the MAGNET adaptation of LLaMA-2-7B taking approximately 7 hours. The hyperparameters for the different training objectives are as follows:

**MNTP.**  We train for 4200 iterations using the Wikitext-103 train set (Merity et al., 2016) with a batch size of 32, a learning rate of 3e-5, and a max sequence length of 512. We select 20% of the tokens for masking – 80% of the selected tokens are replaced with a `[MASK]` token, 10% tokens are replaced with a random token from the model's vocabulary, and 10% tokens are left unchanged. For LLaMA-2-7B, we use "_" as the mask token.

**SSCL.**  We train for 800 iterations with a batch size of 64, a learning rate of 3e-5, and a max sequence length of 128. To extract representations we use the prompt "*Given the sentence, find its representation:*" and extract the representations corresponding to the last token. The training data is created from Wikitext (Merity et al., 2016) by extracting lines longer than 20 words and paraphrasing them for the positive examples. We set $\tau = 0.1$ in equation 3.

**MSG.**  Similar to MNTP, we train for 4200 iterations using the Wikitext-103 train set (Merity et al., 2016) with a batch size of 32, a learning rate of 3e-5, and a max sequence length of 512. A training example can have up to 2 missing spans with span length ranging from 4 to 128 tokens.

**Overall Loss.**  For the first 3400 iterations, we optimize the loss (equation 5) with $\lambda_1 = 1$, $\lambda_2 = 0$, and $\lambda_3 = 1$, and for the next 800 iterations $\lambda_1 = 1$, $\lambda_2 = 9$, and $\lambda_3 = 1$. We initially train with only MNTP and MSG because these objectives help the model learn to capture future context, a capability the base model lacks.

**Word-Level Tasks.**  Using the frozen representations from the last hidden layer of the base model, we train a linear classifier for the three word-level tasks (Chunking, NER, and POS-tagging). Specifically, we train on the CoNLL-2003 train set for 4000 steps using a batch size of 8, a learning rate of 5e-4, and a dropout rate of 0.1.

## B  CONTEXTUAL PROMPT INFILLING

To thoroughly evaluate the infilling capability of the base model, we perform zero-shot and few-shot experiments where the model is shown both preceding and following context of a missing span of text.

### B.1  ZERO-SHOT EVALUATION

To this end, we experimented with four types of prompts to infill a missing line in five-line stories from the ROC Stories dataset. The four prompting strategies we used are:

**Blank Infilling Prompt.**  In this setting, we add a blank token (_) at the infilling position and use the following prompt:

> *Generate the missing line represented by _ in the given text: <text>.*
> *Generate a single sentence.*
> *The missing line is:*

Here, *<text>* represents the input text with "_" in place of a missing sentence.

**Contextual Prompt.**  In this setting, we provide the past and future context of the missing line and use the following prompt:

> *Fill in the missing sentence between "<past-context>" and "<future-context>". Generate only one sentence. The missing sentence is:*

**Prefix-Suffix Prompt.**  In this setting, we give the past context of a missing sentence as a prefix and the future context as a suffix and ask the model to generate the middle. Specifically, we use the following prompt:

*Given the prefix and the suffix, generate the middle sentence.*
*Prefix: <past-context>.*
*Suffix: <future-context>.*
*Generate only one sentence.*
*Middle: .*

**Line-by-Line Prompt.** In this setting, we make the prompt more descriptive by providing all the available context, specifying the line number for all the available lines, and asking for the missing line. For instance, if the task is to infill the second line of a five-sentence story, the prompt would be:

*You have a five-sentence story with some missing text.*
*Here is the context for each line, with the missing line indicated:*
*Line 1: <line-1>*
*Line 2: [Missing Line]*
*Line 3: <line-3>*
*Line 4: <line-4>*
*Line 5: <line-5>*
*Please generate the missing line of the story. Please generate only the missing line and nothing else.*
*The missing line is: Line 2:*

For the abovementioned prompting strategies, we experimented with various prompt variations, including paraphrasing the instructions, using "`[MASK]`", "`[blank]`" or "`_`" to denote the missing line, and addressing common avoidable errors using the instructions (for e.g., adding "*Generate only one line.*" to enforce single line infillings and avoid formatting issues). In general, we find that regardless of the prompting strategy used, LLaMA-2-7B repeats/paraphrases one of the provided lines or summarizes the context as the infilling. In some cases it even ends up generating totally random text (like code). This is perhaps because the model is not trained for the infilling task. Table 9 shows some qualitative examples of text infilling using the different prompting methods.

## B.2 FEW-SHOT EVALUATION

To improve infilling results from the base model, we employed few-shot learning techniques with various prompting styles – blank infilling, prefix-suffix, and line-by-line. Specifically, we provided five solved examples in the model's context using the chosen prompt format and asked the model to infill the missing line in the sixth example. We observed that more descriptive prompts and examples led to better output from the model, and the line-by-line prompting style seemed to be the most effective in enabling coherent infillings. We present qualitative examples of the infilling generated using this approach in Table 10.

## C TRAINING OBJECTIVE ABLATION ANALYSIS

We perform ablation experiments to evaluate the effectiveness of our unified training with the three proposed objectives. Specifically, we compare the performance on representation learning tasks after adapting the LLM using different combinations of the objectives. The results are presented in Table 6. We find that while MNTP is the only objective that explicitly trains the model for better token-level representations, adding MSG marginally improves performance on word-level tasks. We conjecture that MSG, being closer to the original next-token prediction objective of the base LLM, acts as a regularizer and helps prevent extreme variations in the token representations

Table 6: Ablation analysis of the proposed training objectives to assess the potential benefits or downsides of unified training.

| Training Objectives | Chunking | NER | POS | STS12 | STS13 | STS14 | STS15 | STS16 |
|---|---|---|---|---|---|---|---|---|
| MNTP | 92.44 | 98.11 | 93.18 | – | – | – | – | – |
| SSCL | – | – | – | **69.06** | 84.53 | **78.07** | 84.09 | 78.52 |
| MNTP + MSG | 92.51 | 98.20 | **93.38** | – | – | – | – | – |
| SSCL + MSG | – | – | – | 68.46 | 84.52 | 77.33 | **84.35** | 79.17 |
| MNTP + SSCL + MSG | **92.64** | **98.31** | 93.34 | 67.98 | **84.66** | 77.67 | 84.17 | **79.44** |

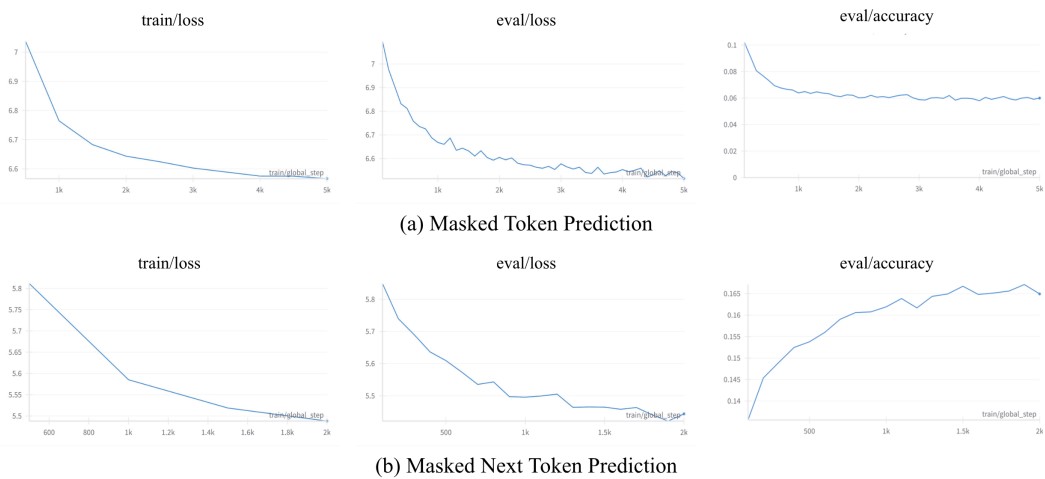

Figure 7: Training curves for MTP and MNTP objectives. When using MTP, model's performance on masked token prediction (measured using eval set accuracy) deteriorates over training iterations.

Table 7: Evaluating the impact of MAGNET on LLaMa-2-7B's performance on the MMLU benchmark.

| Model | Dataset | Humanities | STEM | Social Sciences | Others | Average |
|---|---|---|---|---|---|---|
| LLaMa-2-7B | Unknown | 42.9 | 36.4 | 51.2 | 52.2 | 45.3 |
| MAGNET | Wikitext | 41.1 | 33.5 | 49.5 | 52.1 | 43.7 |
| MAGNET | SlimPajama | 41.7 | 35.8 | 50.9 | 52.8 | 44.9 |

produced by the model. For sentence-level tasks, which use the SSCL objective on the last token's representation, we observe no noticeable benefit or drawback from including MNTP and MSG. This shows that we can add token-level representation learning and infilling capabilities to the model without hampering performance on sentence-level tasks. We conjecture that the effects of unified training on sentence-level tasks are not evident from Table 6 due to the separation of sentence-level representation learning from token-level representation learning and generation, achieved by using only the last token's output as the sentence encoding.

## D    TEXT GENERATION AND KNOWLEDGE RETENTION

In Table 7, we evaluate the impact of MAGNET on the knowledge acquired by the LLM during its pretraining, as measured by the Massive Multitask Language Understanding (MMLU) (Hendrycks et al., 2021) benchmark. We find that adapting the base model using the Wikitext-103 dataset results in a performance decline of 1.6% on average. This degradation can be attributed to the dataset's encyclopedic nature, potentially introducing a knowledge bias that narrows the model's generalization capabilities. To mitigate this limitations, we also adapt the model by fine-tuning on the SlimPajama dataset (Soboleva et al., 2023), which comprises texts from more diverse data sources like Commoncrawl, C4, GitHub, Books, ArXiv, Wikipedia, and StackExchange. With SlimPajama adaptation, the performance drop on MMLU is only 0.4% on average. Some categories (like 'Others') show comparable or even slightly improved performance. This suggests that the adaptation method itself does not have a limiting effect on the model's knowledge and generative ability.

## E    COMPARING MTP AND MNTP OBJECTIVES

Traditionally, language models for representation learning are trained to predict a masked token at position $l$ using the output at position $l$ in the final hidden states (Devlin et al., 2019; Liu et al., 2019; Lan et al., 2019; Sanh et al., 2019). This approach is logical because the residual connections in the transformer block incorporate the $l^{th}$ token's input representation into its output representation.

Table 8: Instructions used for getting sentence encoding for the different sentence-level tasks. "STS*" refers to all the STS tasks.

| Task | Instruction |
|---|---|
| STS* | Retrieve semantically similar text |
| BiorxivClusteringP2P | Identify the main category of Biorxiv papers based on the titles and abstracts |
| BiorxivClusteringS2S | Identify the main category of Biorxiv papers based on the titles |
| TwentyNewsgroupsClustering | Identify the topic or theme of the given news articles |
| MedrxivClusteringP2P | Identify the main category of Medrxiv papers based on the titles and abstracts |
| MedrxivClusteringS2S | Identify the main category of Medrxiv papers based on the titles |

We conducted an experiment to test whether we can use LoRA to adapt the base LLM for $l$-to-$l$ prediction (similar to BERT). The training curves for masked token prediction (MTP) and masked number token prediction (MNTP) are shown in Figure 7. As illustrated, with MTP, the loss converges, but the evaluation accuracy for masked token prediction decreases. This likely occurs because the base model is trained to predict the $(l+1)^{th}$ token at position $l$, and shifting to $l$-to-$l$ prediction introduces a significant distributional shift that the model may struggle to accommodate swiftly. Thus, overall, we find that MNTP is a more effective objective for converting a decoder-only LLM into a text encoder. Additionally, MNTP aligns well with the causal MSG objective and paves way for a unified text generator and encoder.

## F    LIMITATIONS

While MAGNET better preserves the open-ended generation capability of the base LLM compared to other bidirectional adaptation methods, it still reduces generation quality. For instance, fine-tuning LLaMA-2-7B with MAGNET increases the test set perplexity (PPL) on Wikitext-103 from 6.4 to 7.6. Although qualitative analysis shows no major artifacts in the generated text, the model's confidence in predicting the next word seems to be somewhat diminished.

In the infilling task, we only focus on augmenting the base LLM with the ability to consider all the surrounding information to produce contextually coherent infilling. We have observed that the quality of infilling decreases when using the MAGNET-adapted LLM to infill very long sequences in the middle of a text. We believe this observation is a result of the parameters set for the MSG objective, and it can be addressed by the following three ways: (1) Instead of attempting to infill a long segment of say 128 tokens at once, divide it into smaller chunks, such as infill 32 tokens at a time; (2) Adjusting the parameters of the MSG task; (3) Fine-tuning the model specifically for infilling tasks.

Since the pre-training data for LLaMA-2-7B is not publicly available, there is a potential risk of test set contamination. To this end, we compare test set performances with the original LLaMA-2-7B and other recent methods that use the same base model (LLM2Vec and Echo Embeddings). This comparative analysis helps ensure that our results are not unduly influenced by data contamination. We leave it to the future works to establish and test on benchmarks that are surely not a part of the pre-training data of the LLMs.

Table 9: Zero-shot sentence infilling examples using Llama-2-7B with different prompting strategies.

| Method | Infilled Line |
|---|---|
| *John was at the gym.* �_ *He couldn't get the barbell off his chest. He was too embarrassed to call for help. John suffocated as the weights crushed his chest.* | |
| **Blank Infilling** | John was at the gym. |
| **Contextual** | John was at the gym, trying to lift a particularly heavy barbell, when he suddenly found himself unable to breathe and his face turned blue. |
| **Prefix-Suffix** | John suffocated as the weights crushed his chest. |
| **Line-by-Line** | John was at the gym, sweating profusely. |
| *Toni took a trip to the local fair with her family.* ▓ *The balloon was red and star shaped. Unfortunately it slipped out of her hands. She was sad to see it go.* | |
| **Blank Infilling** | " She watched as it soared high up in the sky."tt = 0 while t < 10: t += 0. |
| **Contextual** | Toni took a trip to the local fair with her family and won a red and star-shaped balloon as a prize. |
| **Prefix-Suffix** | Toni took a trip to the local fair with her family and was excited to see a red star-shaped balloon. |
| **Line-by-Line** | Toni took a trip to the local fair with her family. |
| *I was resting on my couch.* ▓ *They raised their water guns. I immediately ran. I went outside and sprayed them with the hose when they followed.* | |
| **Blank Infilling** | They raised their water guns at me. |
| **Contextual** | I heard the sound of water splashing and knew they were coming for me. |
| **Prefix-Suffix** | I immediately ran. |
| **Line-by-Line** | I was resting on my couch, bored out of my mind. |
| *Kaylee owned a ferret who always ran free around the house.* ▓ *She put flyers all over the neighborhood. There was a knock at the door three days later. Someone found her ferret.* | |
| **Blank Infilling** | She was overjoyed to see her ferret wagging its tail when she opened the door. |
| **Contextual** | Kaylee put flyers all over the neighborhood. |
| **Prefix-Suffix** | Kaylee found her ferret three days after putting flyers all over the neighborhood. |
| **Line-by-Line** | She found the ferret hiding under the bed. |
| *Cody really wanted a smart phone.* ▓ *Cody protested that all of his friends had their own phones. His parents phoned some of the other parents. His parents learned that Cody had been exaggerating.* | |
| **Blank Infilling** | Cody really wanted a smart phone. |
| **Contextual** | Cody's parents discovered that he had been lying about wanting a smartphone when they phoned some of the other parents and found out that none of their children had ever mentioned anything about Cody wanting a new phone. |
| **Prefix-Suffix** | Cody's parents phoned some of the other parents. |
| **Line-by-Line** | Cody really wanted a smart phone because he thought it would make him cooler. |
| *Darla was afraid of spiders.* ▓ *She screamed and jumped away. She yelled for her father to come in and help. Her father came and removed the spider.* | |
| **Blank Infilling** | Darla was afraid of spiders. |
| **Contextual** | Darla was afraid of spiders because she had a bad experience with one when she was younger. |
| **Prefix-Suffix** | Darla was afraid of spiders and screamed and jumped away. |
| **Line-by-Line** | Darla was afraid of spiders. |

Table 10: Sentence infilling examples using LLaMA-2-7B and its adaptation using MAGNET. Uni-Llama refers to the unidirectional model that only considers the left context, and FS-Llama is the few-shot variant that learns to use the full context to generate the infilling.

| Method | Infilled Line |
|---|---|
| *John was at the gym.* �____ *He couldn't get the barbell off his chest. He was too embarrassed to call for help. John suffocated as the weights crushed his chest.* | |
| **Uni-Llama** | He was on the treadmill, running at a steady pace. |
| **FS-Llama** | He was doing bench presses. |
| **MAGNET** | John was doing bench presses when he accidentally dropped the barbell on top. |
| *Toni took a trip to the local fair with her family.* �____ *The balloon was red and star shaped. Unfortunately it slipped out of her hands. She was sad to see it go.* | |
| **Uni-Llama** | She won a prize for her drawing of a cat. |
| **FS-Llama** | She won it in a game of darts. |
| **MAGNET** | She was so excited to see the balloon vendor and bought a red one. |
| *I was resting on my couch.* �____ *They raised their water guns. I immediately ran. I went outside and sprayed them with the hose when they followed.* | |
| **Uni-Llama** | I was feeling a bit under the weather, so I decided to take a break. |
| **FS-Llama** | My neighbor's kids were playing tag in the backyard. |
| **MAGNET** | I heard a noise and looked out the window and saw two boys with water guns. |
| *Kaylee owned a ferret who always ran free around the house.* �____ *She put flyers all over the neighborhood. There was a knock at the door three days later. Someone found her ferret.* | |
| **Uni-Llama** | She loved her ferret, but she knew that ferrets have a tendency to get into things they shouldn't. |
| **FS-Llama** | She put up a reward for the safe return of her ferret. |
| **MAGNET** | She was devastated when she lost him and searched everywhere for him. |
| *Cody really wanted a smart phone.* �____ *Cody protested that all of his friends had their own phones. His parents phoned some of the other parents. His parents learned that Cody had been exaggerating.* | |
| **Uni-Llama** | He had been saving up his allowance for months, but he just couldn't. |
| **FS-Llama** | Cody's friends told him they had been saving up for their phones. |
| **MAGNET** | His parents refused to buy him one because they thought it was too expensive and dangerous. |
| *Darla was afraid of spiders.* �____ *She screamed and jumped away. She yelled for her father to come in and help. Her father came and removed the spider.* | |
| **Uni-Llama** | She had always been afraid of them, even as a child. |
| **FS-Llama** | She saw a spider crawling on her bed. |
| **MAGNET** | She was in her room when she saw a spider on her bed. |
| *A strange structure appeared on the outside of Connie's house one day.* �____ *She called an exterminator to come get them. The exterminator came and got rid of the wasps quickly. Connie was happy that the nest was gone.* | |
| **Uni-Llama** | It was a large, glowing crystal that seemed to pulse with. |
| **FS-Llama** | She saw a wasp flying around her kitchen window. |
| **MAGNET** | Connie noticed that there were wasps in the house and outside. |

Table 11: Text generated using greedy decoding with BERT-Base, LLaMA-2-7B, and its adaptations using LLM2Vec and MAGNET. To generate text from BERT, we recursively add a mask token at the end of the input sentence and use the model to predict the mask.

| Model | Generated Text |
|---|---|
| | Initial phrase: *The film was well received* |
| Llama | *The film was well received* by critics and audiences alike, with many praising its unique blend of humor and heart. Here are some of the key reasons why critics and audiences loved "The Big Sick": 1. Unique blend of humor and heart: The film's ability to balance humor and heart was widely praised by critics and audiences. The film's lighthearted moments are often juxtaposed with more serious and emotional scenes, creating a film that is both funny and heartwarming. 2. Authentic representation: The film is ... |
| LLM2Vec | *The film was well received* by critics and audiences alike. The film was well received by critics and audiences alike. The film was well received by critics and audiences alike. The film was well received by critics and audiences alike. The film was well received by critics and audiences alike. The film was well received by critics and audiences alike. The film was well received by critics and audiences alike. The film was well received by critics and ... |
| MAGNET | *The film was well received* by critics and audiences alike and was nominated for several awards including the Academy Award for Best Visual Effects and the Saturn Award for Best Science Fiction Film. The film was also a commercial success and grossed over $100 million at the box office. = = Plot = = In the year 2018, a meteorite crashes into the Pacific Ocean, causing a massive tsunami that destroys most of the world's coastal cities. The survivors of the disaster band together ... |
| BERT | *The film was well received* and the " " " " " " " " " " " " " " are " " are " are " are " are are " are are the are the the the the the the the the the the the the the so so and and and and and and and and and and and and and and and and and and and and and and and and and and and and and and and and and and and and and and and and ... |
| | Initial phrase: *After 1805, the exploits of* |
| Llama | *After 1805, the exploits of* the British East India Company in India became increasingly controversial, and the company faced growing criticism from both the British public and Indian nationalists. Here are some of the key events that contributed to the decline of the British East India Company: 1. The Battle of Buxar (1757): The British East India Company defeated the French and their Indian allies in the Battle of Buxar, which gave them control over Bengal. This marked the beginning of the company's expansion in India. 2. The Battle of Plassey ... |
| LLM2Vec | *After 1805, the exploits of* the French army in Egypt continued to be successful. In 1806, the French army defeated the Ottoman army at the Battle of Aboukir. In 1807, the French army defeated the Ottoman army at the Battle of Austerlitz. In 1808, the French army defeated the Ottoman army at the Battle of Wadi Abu Jarai. In 1809, the French army defeated the Ottoman army at the Battle of Roli. In 1810, the French army defeated ... |
| MAGNET | *After 1805, the exploits of* the pirates of the Caribbean became less frequent and the British Navy began to take a more active role in suppressing piracy. The British Navy began to patrol the Caribbean and the Spanish Main, and the pirates were forced to flee to other areas. The pirates of the Caribbean were not completely eliminated, however, and piracy continued to be a problem in the Caribbean until the mid 19th century. The last pirate to be executed in the Caribbean was Captain Kidd, who was hanged ... |
| BERT | *After 1805, the exploits of* the the - the - way - to - the - other - to - the - other - to - the - other - to - the - other -. -. -. -. -. -. -. -. -. -. -. -. -. -. -. -. -. -. - " - " - " " - " " " " " " " " " " " " - " " - " " " " " " " " " " and the " - " " " - " " " - " - " " - " " " - " " " " " " " " |

