# OpenReview forum: "MAGNET: Augmenting Generative Decoders with Representation Learning and Infilling Capabilities"
_ICLR.cc/2025/Conference — Submitted to ICLR 2025_

### Official Review · Reviewer_Ygdb · 2024-11-01

**Soundness:** 3
**Presentation:** 3
**Contribution:** 2
**Rating:** 5
**Confidence:** 4

**Summary:**

This work introduced MAGNET ((Modified Attention for Generation and Encoding of Text), a method to transform causal LLMs into text encoders and infilling language models with bidirectional context capturing ability.
Three training objectives are introduced including: masked next token prediction, sentence level contrastive learning, and missing span generation,  to adapt the LLM to different tasks.
Experimental results shows a significant improvement in sequence labeling and text-filling tasks over the baseline
and a fair performance in open-ended text generation.

**Strengths:**

1. I think the idea to combine both unidirectional and bidirectional attention is new. This enhanced the representation learning and infilling capabilities of LLMs, while retaining their core generative functions.
2. I like the identification of token repetition problem in open-text generation, and the proposed method does have a positive impact in this direction.

**Weaknesses:**

1.  I am a little confused of the motivation. The author try to design a unified framework for text generation and text encoding,
by introducing some training objectives in bidirectional models (BERT, ERNIE) to LLMs.    The resulting model could do both tasks with a fair performance. But what is the purpose of the unified framework ? Could each task benefit from each other? I don't see much evidence in this paper.

2. The proposed three training objectives are not new: mask next token prediction is a variant task of LLMs, and I think MNTP is not a proper name, since the token predicted is actually not the "next" one, with the bidirectional setting.
SSCL is widely studied in representation learning, such as <On learning universal representations across languages> ICLR21.
And the MSG is a similar version of ERNIE (Enhanced Language Representation with Informative Entities).
And I suggest the author to explore more sophisticated loss function, rather than a simple linear combination of these three.
So in general I think the innovation of this paper is limited.

3. For the experiment part, I suggest the author to compare with stronger baseline on word-level and STS tasks like XLNet and StructBERT, since BERT and RoBERT(2019) is a little out-dated.

**Questions:**

1. How much does the adaptation of MAGNET hurt the text generation ability of LLMs ?  I think if the author claims to potentially unify text generation and text encoding within a single framework. This question should be carefully studied.

2. How does the proposed model compare with the SOTA performance in STS and word-level tasks?

---

> ### Author Response · Authors · 2024-11-20
> **Response to the Official Review of Submission3348 by Reviewer Ygdb (1/2)**
>
> Thank you for thoroughly reviewing our work. We are glad to hear that you appreciated the deployed attention mechanism and our analysis of the repetition problem. We have tried to address each of your concerns point by point in this rebuttal.
>
>
> > (Weakness 1) What is the purpose of the unified framework? Could each task benefit from each other?
>
> **Ans.** In Appendix C of our paper, we perform an ablation analysis demonstrating the impact of unified training on the model's performance on the representation learning tasks. Notably, we observe that while token-level representation learning is purely promoted by the MNTP loss, adding MSG (which is a generative loss) boosts the results across all token-level classification tasks.
>
> Secondly, the results in Tables 1, 2, and 3, demonstrate the superior performance of our approach over LLM2Vec and LLM2Vecᴹᴺᵀᴾ [1]. This also underscores the beneficial interaction between different objectives in a unified framework. This is because LLM2Vecᴹᴺᵀᴾ uses the same MNTP objective as our approach, but in isolation. By simultaneously training for multiple objectives, we demonstrate superior performance on the representation learning tasks.
>
> Besides the beneficial synergy between losses, training unified models for multiple tasks offers compelling practical and performance benefits too. For instance, these models require less storage and computational overhead compared to deploying multiple specialized models. Training a unified model is also more resource-efficient and it can ensure consistency across related tasks, which is crucial for real-world applications where multiple capabilities must work together seamlessly. Thus, owing to these benefits, our goal in this work was to explore whether and how a single LLM can effectively perform representation learning and text generation (infilling and open-ended).
>
> We acknowledge that these points could have been emphasized more effectively. As a result, we have revised our draft to highlight them more clearly, with the most significant updates in Sections 4.1 and 4.2.
>
>
> > (Weakness 2) The proposed three training objectives are not new. I suggest the author to explore a more sophisticated loss function, rather than a simple linear combination of these three.
>
> **Ans.** We do not claim the novelty of the loss function itself. In our view, the innovation of the paper lies in combining three widely-used self-supervised loss functions for language modeling with a nuanced attention mechanism, demonstrating the possibility of a unified model that can perform both representation learning and text infilling. To the best of our knowledge, no prior work has attempted to simultaneously augment a pretrained LLM with these capabilities and evaluate them within a single framework.
>
> We adopt the formulation and naming of the MNTP task as described in [1]. Appendix E provides some justification for this naming convention and explores the potential application of the standard MTP objective for token-level representation learning using decoder-only models.
>
>
> > (Weakness 3 and Question 2) Compare with stronger baselines on word-level and STS tasks like XLNet and StructBERT.
>
> **Ans.** We present a performance comparison between the proposed approach, XLNet, and StructBERT in the table below.
>
> | Model      | Chunking | NER   | POS-Tag | STS12 | STS13 | STS14 | STS15 | STS16 |
> |------------|----------|-------|---------|-------|-------|-------|-------|-------|
> | XLNet      | 79.70    | 93.67 | 83.02   | 5.91  | 4.78  | 0.83  | 8.14  | 13.63 |
> | StructBERT| 89.99    | 97.31 | 90.86   | 20.83 | 38.84 | 24.42 | 40.39 | 59.75 |
> | MAGNET     | 92.64    | 98.31 | 93.34   | 67.98 | 84.66 | 77.67 | 84.17 | 79.44 |
>
> Following the approach outlined in the paper, we train a linear classifier for token-level tasks and perform zero-shot evaluation on sentence-level tasks. We observe that LLaMa-2-7B, adapted using our method, outperforms the text encoders across all tasks by a significant margin.
>
> For XLNet, we experiment with both the last token and mean pooled token representations and report the best results. For StructBERT, we use the CLS token representation. The results on the STS tasks demonstrate that these strong encoder models do not perform well for zero-shot sentence-level tasks and require additional fine-tuning for effective performance. In contrast, LLMs adapted using MAGNET can effectively handle these tasks. Additionally, fine-tuning XLNet and StructBERT with contrastive loss may negatively impact their performance on word-level tasks, whereas our method offers a unified framework that maintains strong performance across both word- and sentence-level tasks.
>
>
> [1] BehnamGhader et al., LLM2Vec: Large Language Models Are Secretly Powerful Text Encoders, COLM 2024
>
>
> *continued*

---

> > ### Author Response · Authors · 2024-11-20
> > **Response to the Official Review of Submission3348 by Reviewer Ygdb (2/2)**
> >
> > > (Question 1) How much does the adaptation of MAGNET hurt the text generation ability of LLMs? I think if the author claims to potentially unify text generation and text encoding within a single framework, this question should be carefully studied.
> >
> > **Ans.** Based on our evaluations, MAGNET has minimal impact on the text generation ability of the base LLM. In the paper, we demonstrate this in the following ways:
> > 1. We demonstrate that, compared to the base LLM, the adapted LLM exhibits only a slight increase in the tendency to repeat words and phrases. This performance is several orders of magnitude better than other methods capable of encoding text. (Section 4.4).
> > 2. Through human evaluation, we show that our method enables the adapted LLM to perform text infilling—an additional capability that emerges from our adaptation. In contrast, the base model performs this task with limited effectiveness. (Section 4.3).
> > 3. We present qualitative examples in Table 11, showcasing text generated from a seed prompt by our model and the baselines. These examples illustrate that, similar to the base LLM, the adapted LLMs produce coherent text with proper grammar and logical structure.
> >
> > Further, we perform additional experiments to show that after adaptation using MAGNET, the model retains the knowledge and generation capabilities of the base model. Specifically, we conduct evaluations using the MMLU benchmark to compare our adapted model against the base LLaMa-2-7B model by fine-tuning on two datasets:
> >
> > Wikitext: The post-adaptation results we present in the paper use the Wikitext-103 dataset. When the base model is adapted using only the Wikipedia data, its performance drops by 1.6% on average. This degradation can be attributed to the dataset's encyclopedic nature, potentially introducing a knowledge bias that narrows the model's generalization capabilities.
> >
> > SlimPajama: To mitigate the limitations observed in the Wikitext-based adaptation, we also adapt the model by fine-tuning on the SlimPajama dataset, which comprises a mixture of texts from more diverse data sources like Commoncrawl, C4, GitHub, Books, ArXiv, Wikipedia, and StackExchange. Notably, using this dataset, the performance drop on MMLU is only 0.4% on average, with some categories (like 'Others') showing comparable or even slightly improved performance.
> >
> > This suggests that the adaptation method itself does not have a limiting effect on the model's knowledge and generative ability.
> >
> > |     | Dataset    | Humanities    | STEM    | Social Sciences    | Others    | Average    |
> > |-------------|-------------|-------------|-------------|-------------|-------------|-------------|
> > | LLaMa-2-7B | Unknown    | 42.9 | 36.4 | 51.2 | 52.2 | 45.3 |
> > | MAGNET | Wikitext    | 41.1 | 33.5 | 49.5 | 52.1 | 43.7 |
> > | MAGNET | SlimPajama    | 41.7 | 35.8 | 50.9 | 52.8 | 44.9 |
> >
> > (We have included the MMLU experiment in the revised version of our paper)

---

> > > ### Author Response · Authors · 2024-11-23
> > > **Request to Review the Rebuttal**
> > >
> > > Dear Reviewer Ygdb,
> > >
> > > Thank you for providing valuable feedback on our submission. We have submitted our response to your concerns, along with a revised version of the paper. With only a few days remaining until the rebuttal deadline, we would greatly appreciate it if you could review our response and let us know if you have any additional questions or concerns. We are eager to ensure all your points are thoroughly addressed and look forward to your feedback.
> > >
> > > Best regards,
> > >
> > > Authors of Submission3348

---

> > > > ### Author Response · Authors · 2024-11-25
> > > > **Sincerely Awaiting Your Feedback**
> > > >
> > > > Dear Reviewer Ygdb,
> > > >
> > > > As the discussion period concludes tomorrow, we kindly request you to review our response and share any remaining questions or concerns. We are committed to addressing all your points comprehensively and look forward to your valuable feedback.
> > > >
> > > > Best regards

---

> > > > > ### Author Response · Authors · 2024-11-27
> > > > > **Follow-Up Request for Feedback**
> > > > >
> > > > > Dear Reviewer Ygdb,
> > > > >
> > > > > With the deadline to update the paper approaching in a few hours, we wanted to kindly follow up to see if our response addresses your concerns. If you have any remaining questions or suggestions, please let us know—we greatly value your feedback and are happy to address any additional points.
> > > > >
> > > > > Thank you for your time.
> > > > >
> > > > > Best regards

---

> ### Author Response · Authors · 2024-12-01
> **Follow-up on Reviewer Feedback**
>
> Dear Reviewer Ygdb,
>
> We are following up on our previous responses addressing your concerns. We are eager to understand if our revisions meet your requirements or if you need any additional clarifications. Your specific feedback will help us finalize and improve our submission.
>
> We look forward to your reply.

---

### Official Review · Reviewer_qxxH · 2024-11-05

**Soundness:** 2
**Presentation:** 3
**Contribution:** 2
**Rating:** 3
**Confidence:** 5

**Summary:**

The paper introduces MAGNET, a novel adaptation of decoder-only large language models (LLMs) designed to enhance both representation learning and text infilling capabilities while maintaining their original generative functions. Traditionally, unidirectional and bidirectional models are trained separately with distinct objectives—either generation or representation learning—thus missing the potential benefits of integrating these objectives. MAGNET addresses this gap by employing three self-supervised training objectives and introducing an attention mechanism that combines bidirectional and causal attention. This unified approach allows for simultaneous training across all objectives. The results demonstrate that LLMs adapted with MAGNET outperform state-of-the-art text encoders in token-level and sentence-level representation learning tasks. Additionally, MAGNET enhances the base LLM's ability to generate contextually appropriate text infillings by considering future context. Unlike other bidirectional models focused solely on representation learning, MAGNET-adapted LLMs retain the ability to perform open-ended text generation.

**Strengths:**

1. The proposed method enhances both representation learning and text infilling capabilities while preserving the original generation ability through a multi-level training objective.

2. Experimental results demonstrate that the proposed MAGNET outperforms traditional text encoders and decoders. Specifically, LLMs adapted with MAGNET surpass state-of-the-art text encoders in token-level and sentence-level representation learning tasks.

**Weaknesses:**

1. Several works have aimed to optimize pre-trained models for both generation and understanding tasks, such as XLNet, ERNIE, and GLM. The authors should provide a comparison with these approaches.

2. Contrastive loss has been widely employed in representation learning, as demonstrated by Gunel et al. (2022) and Wei et al. (2021). The authors should present a comprehensive related work section and compare relevant studies to highlight the effectiveness of their proposed approach.

Gunel et al., 2022. Supervised contrastive learning for pre-trained language model fine-tuning
Wei et al., 2021. On learning universal representations across languages

**Questions:**

N/A

---

> ### Author Response · Authors · 2024-11-20
> **Response to the Official Review of Submission3348 by Reviewer qxxH**
>
> Thank you for taking the time to review our work. We appreciate your constructive suggestions and have made efforts to address them in this response.
>
>
> > Regarding comparison with other unified pre-training approaches
>
> We compare the model adapted using our approach with XLNet on word-level representation learning tasks, sentence-level representation learning tasks, and a text generation task.
>
> - Word-level representation learning: Following the approach outlined in the paper, we train a linear classifier for token-level tasks of Chunking, NER, and POS-tagging. We find that our approach outperforms XLNet on all these tasks.
>
> | Model      | Chunking | NER   | POS-Tag |
> |------------|----------|-------|---------|
> | XLNet      | 79.70    | 93.67 | 83.02   |
> | MAGNET     | 92.64    | 98.31 | 93.34   |
>
> - Sentence-level representation learning: We perform zero-shot evaluations on the STS tasks. For XLNet, we evaluated both the last token and mean pooling strategies for generating sentence encoding and selected the best of two numbers for comparison. The results demonstrate that XLNET does not perform well for zero-shot sentence-level tasks and requires additional fine-tuning for effective performance.
>
> | Model      | STS12  | STS13  | STS14  | STS15  | STS16  |
> |------------|--------|--------|--------|--------|--------|
> | XLNet      | 5.91   | 4.78   | 0.83   | 8.14   | 13.63  |
> | MAGNET     | 67.98  | 84.66  | 77.67  | 84.17  | 79.44  |
>
>
> - Text generation: We also evaluate XLNet on the MMLU benchmark, and our evaluation reveals severe limitations in its ability to handle the formal evaluation setup for such tasks. Specifically, when presented with the standard input format of questions and multiple choices, XLNet consistently fails to generate responses in the required format (which is not the case with our model). To ensure a thorough assessment, we also attempt an alternative evaluation approach by analyzing the model's prediction scores for each option and selecting the highest-scoring choice. Even with this modified methodology, XLNet-Large achieves only 21.4% accuracy compared to our approach's 43.7%. These results underscore XLNet's fundamental limitations in two critical areas: the inability to generate responses in the required format, and insufficient language understanding/reasoning capability to perform effectively on comprehensive benchmarks like MMLU.
>
> While our evaluation of XLNet provides useful insights, we acknowledge that it may not be the most relevant baseline given our research objectives. Our work focuses on augmenting powerful LLMs with additional capabilities such as representation learning and infilling, rather than developing a new pre-training framework from scratch. Therefore, we believe that more appropriate comparisons would be with other adaptation methods that modify pre-trained LLMs for specific tasks, as well as with the base LLMs that we adapt. These baselines would better align with our goal of enhancing, rather than replacing, existing language models.
>
>
> > The authors should present a comprehensive related work section.
>
> Thank you for the feedback! In the revised version, we have significantly expanded our related work section to better position our work in the context of the existing approaches that unify language understanding and generation.

---

> > ### Author Response · Authors · 2024-11-23
> > **Request to Review the Rebuttal**
> >
> > Dear Reviewer qxxH,
> >
> > Thank you for providing valuable feedback on our submission. We have submitted our response to your concerns, along with a revised version of the paper. With only a few days remaining until the rebuttal deadline, we would greatly appreciate it if you could review our response and let us know if you have any additional questions or concerns. We are eager to ensure all your points are thoroughly addressed and look forward to your feedback.
> >
> > Best regards,
> >
> > Authors of Submission3348

---

> > > ### Author Response · Authors · 2024-11-25
> > > **Sincerely Awaiting Your Feedback**
> > >
> > > Dear Reviewer qxxH,
> > >
> > > As the discussion period concludes tomorrow, we kindly request you to review our response and share any remaining questions or concerns. We are committed to addressing all your points comprehensively and look forward to your valuable feedback.
> > >
> > > Best regards

---

> > > > ### Author Response · Authors · 2024-11-27
> > > > **Follow-Up Request for Feedback**
> > > >
> > > > Dear Reviewer qxxH,
> > > >
> > > > With the deadline to update the paper approaching in a few hours, we wanted to kindly follow up to see if our response addresses your concerns. If you have any remaining questions or suggestions, please let us know—we greatly value your feedback and are happy to address any additional points.
> > > >
> > > > Thank you for your time.
> > > >
> > > > Best regards

---

> ### Author Response · Authors · 2024-12-01
> **Follow-up on Reviewer Feedback**
>
> Dear Reviewer qxxH,
>
> We are following up on our previous responses addressing your concerns. We are eager to understand if our revisions meet your requirements or if you need any additional clarifications. Your specific feedback will help us finalize and improve our submission.
>
> We look forward to your reply.

---

### Official Review · Reviewer_Ashd · 2024-11-06

**Soundness:** 3
**Presentation:** 3
**Contribution:** 3
**Rating:** 6
**Confidence:** 3

**Summary:**

This paper introduces MAGNET, a model adaptation framework aimed at enhancing decoder-only large language models (LLMs) with representation learning and infilling capabilities, without compromising their generative performance. MAGNET employs a modified attention mechanism that integrates bidirectional and causal attention, supporting unified training across three self-supervised objectives: masked next token prediction, contrastive learning, and missing-span generation. Experimental results show that MAGNET surpasses state-of-the-art models in token-level and sentence-level representation tasks and excels in text infilling, making it a versatile approach for various natural language processing (NLP) tasks.

**Strengths:**

1. Proposes an innovative modification to the attention mechanism of LLMs, balancing bidirectional and causal attentions for improved versatility.
2. Demonstrates strong empirical results, showing MAGNET's effectiveness in multiple tasks, including representation learning and text infilling.
3. Provides a comprehensive analysis of MAGNET’s performance compared to state-of-the-art models, highlighting improvements in both quality and efficiency.

**Weaknesses:**

The main problem is the lack of experiments on text generation task. Text generation is the most important task since any NLP tasks can be transformed into the format of text generation, so it might be undesirable if the method sacrifices the text generation ability for text understanding ability. The paper only studied the text repetition problem of the method, but did not test it on widely-used benchmarks for LLMs evaluation.

**Questions:**

Have you tried other method of using Llama features? For example, using pooled embeddings? The reason I ask is because decoder model is not explicitly trained for text understanding, it is unclear what is the best way to utilize them for text understanding task.

---

> ### Author Response · Authors · 2024-11-20
> **Response to the Official Review of Submission3348 by Reviewer Ashd**
>
> Thank you for your thorough review of our work. We are glad that you found the proposed approach innovative and most of the experiments satisfactory. Below, we address the weaknesses and questions you raised.
>
> > (Weakness) Lack of experiments on text generation tasks.
>
> **Ans.** Thank you for bringing this to our attention! To substantiate the model's text generation capabilities post-adaptation, we conduct evaluations using the MMLU benchmark. We compare our adapted model against the base LLaMa-2-7B model by fine-tuning on two datasets:
>
> - Wikitext: The post-adaptation results we present in the paper use the Wikitext-103 dataset. When the base model is adapted using only the Wikipedia data, its performance drops by 1.6% on average. This degradation can be attributed to the dataset's encyclopedic nature, potentially introducing a knowledge bias that narrows the model's generalization capabilities.
>
> - SlimPajama: To mitigate the limitations observed in the Wikitext-based adaptation, we also adapt the model by fine-tuning on the SlimPajama dataset, which comprises a mixture of texts from more diverse data sources like Commoncrawl, C4, GitHub, Books, ArXiv, Wikipedia, and StackExchange. Notably, using this dataset, the performance drop on MMLU is only 0.4% on average, with some categories (like 'Others') showing comparable or even slightly improved performance.
>
>
> |     | Dataset    | Humanities    | STEM    | Social Sciences    | Others    | Average    |
> |-------------|-------------|-------------|-------------|-------------|-------------|-------------|
> | LLaMa-2-7B | Unknown    | 42.9 | 36.4 | 51.2 | 52.2 | 45.3 |
> | MAGNET | Wikitext    | 41.1 | 33.5 | 49.5 | 52.1 | 43.7 |
> | MAGNET | SlimPajama    | 41.7 | 35.8 | 50.9 | 52.8 | 44.9 |
>
> So, all in all, we demonstrate the text generation capabilities of the adapted model using the following:
> 1. We show that the model retains the knowledge and generation capabilities of the base model through experiments on the MMLU benchmark (as discussed above).
> 2. We show that, compared to other text encoders (like BERT) and LLM-adaptation methods (like LLM2Vec), the models adapted using our approach are significantly less prone to repeating words and phrases  (Section 4.4).
> 3. Through human evaluation, we demonstrate that our method adapts LLM to effectively perform text infilling, which is inherently a generative task (Section 4.3).
> 4. We also include qualitative examples in Table 11 of the paper, showcasing the text generated from a seed prompt by our model and the baselines.
>
>
> > (Question) Have you tried other methods of using Llama features (for sentence encoding)?
>
> **Ans.** We compare different strategies for using the LLaMa-2-7B features for the STS task. Specifically, we evaluate three approaches for encoding a sentence: (1) mean pooling across all token representations, (2) using the last token representation, and (3) introducing a learnable CLS token. We use SimCSE fine-tuning to assess each pooling strategy's effectiveness in generating meaningful sentence embeddings, and the results are presented in the table below.
>
> |             | STS12   | STS13   | STS14   | STS15   | STS16   | STS17   | STS22   | Avg     |
> |--------------------|---------|---------|---------|---------|---------|---------|---------|---------|
> | Mean Pool          | 60.02   | 77.26   | 70.22   | 80.06   | 79.63   | 83.94   | 59.86   | 73.00   |
> | EOS Token          | 69.75   | 74.63   | 67.27   | 78.69   | 77.65   | 84.69   | 58.96   | 73.09   |
> | Learned CLS Token  | 59.39   | 71.02   | 66.78   | 78.01   | 75.48   | 63.20   | 29.45   | 63.33   |
>
> We find that the last token and mean pooling produce comparable results, outperforming the learned CLS token. In our setup, we prefer using the last token representation, as it enables the separation of tokens used for word-level and sentence-level representation learning tasks (please see Sections 3.2.1 and 3.2.2 for more details).

---

> > ### Author Response · Authors · 2024-11-23
> > **Request to Review the Rebuttal**
> >
> > Dear Reviewer Ashd,
> >
> > Thank you for providing valuable feedback on our submission. We have submitted our response to your concerns, along with a revised version of the paper. With only a few days remaining until the rebuttal deadline, we would greatly appreciate it if you could review our response and let us know if you have any additional questions or concerns. We are eager to ensure all your points are thoroughly addressed and look forward to your feedback.
> >
> > Best regards,
> >
> > Authors of Submission3348

---

> > > ### Author Response · Authors · 2024-11-25
> > > **Sincerely Awaiting Your Feedback**
> > >
> > > Dear Reviewer Ashd,
> > >
> > > As the discussion period concludes tomorrow, we kindly request you to review our response and share any remaining questions or concerns. We are committed to addressing all your points comprehensively and look forward to your valuable feedback.
> > >
> > > Best regards

---

> > > > ### Comment · Reviewer_Ashd · 2024-11-25
> > > >
> > > > Thanks for the new results. Are MMLU experiments conducted on all of the categories or just a subset? Seems that the score of LLama-2-7B is different what is shown on the leaderboard: https://crfm.stanford.edu/helm/mmlu/latest/#/leaderboard. However, I am still willing to improve my score.

---

> > > > > ### Author Response · Authors · 2024-11-26
> > > > > **Thank You!**
> > > > >
> > > > > We are pleased to have addressed your concerns and sincerely thank you for reviewing our work and response. Your insightful comments have helped enhance the quality of our paper.
> > > > >
> > > > > Regarding your question: We report results across all categories of the MMLU test set, following the same evaluation approach as used in the HELM framework. The only difference we identified between the HELM evaluation and our setup is in the ordering of the five in-context examples. In our setup, we construct the in-context prompt using the dev set of `cais/mmlu` from `load_dataset()`, maintaining the order in which the examples are retrieved by this function. This ordering appears to differ from the one used in the HELM evaluation.
> > > > > For instance, the ordering of examples in the abstract algebra category differs between [Hugging Face](https://huggingface.co/datasets/cais/mmlu/viewer/abstract_algebra/dev) and [HELM](https://crfm.stanford.edu/helm/mmlu/latest/#/runs/mmlu:subject=abstract_algebra,method=multiple_choice_joint,model=meta_llama-2-7b,eval_split=test,groups=mmlu_abstract_algebra), where the HELM's in-context example ordering can be viewed by clicking "Request Details" on the linked page.

---

### Official Review · Reviewer_V5cp · 2024-11-13

**Soundness:** 2
**Presentation:** 3
**Contribution:** 3
**Rating:** 5
**Confidence:** 3

**Summary:**

This paper introduces MAGNET, a decoder-only LLM enhanced in its ability to generate robust representations and infilling missing text spans while preserving its original text generation capabilities. The model's training integrates both unidirectional and bidirectional attention mechanisms. It is also trained with three objectives: masked next token prediction, self-supervised contrastive learning, and missing span generation.

**Strengths:**

* The paper is very well written.
* The concept of a single model that combines the strengths of both masked language models and causal language models is interesting.

**Weaknesses:**

* **Claims with Insufficient Experimental Results** My main concern is that the authors claim the model retains its original text generation capabilities, but they provide insufficient experimental results to support this. Maybe the authors can demonstrate some results on commonly used text generation tasks, e.g. natural language understanding (MMLU, BigBench), summarization.

* **Lack of Motivation** It's unclear whether the results reported in Section 4 reflect zero-shot performance or task-specific fine-tuning. I believe zero-shot performance would be more meaningful, as it would more convincingly demonstrate the effectiveness of a unified model in both semantic representation and text generation tasks. Otherwise, selecting the optimal attention mechanism and training objective for each specific task would be more practical.

**Questions:**

In the "Overall Loss" section of Appendix A, it's unclear why a two-stage training approach is necessary and why only two objectives are used in the first stage.

---

> ### Author Response · Authors · 2024-11-20
> **Response to the Official Review of Submission3348 by Reviewer V5cp**
>
> Thank you for your thorough review. We are pleased you appreciated the writing and the proposed concept. We value your constructive feedback and address your concerns point-by-point in this response.
>
>
> > (Weakness 1) Substantiating the claim that the model retains its text generation capabilities post-adaptation with our approach.
>
> **Ans.** To substantiate the model's text generation capabilities post-adaptation, we conduct evaluations using the MMLU benchmark. We compare our adapted model against the base LLaMa-2-7B model by fine-tuning on two datasets:
>
> - Wikitext: The post-adaptation results we present in the paper use the Wikitext-103 dataset. When the base model is adapted using only the Wikipedia data, its performance drops by 1.6% on average. This degradation can be attributed to the dataset's encyclopedic nature, potentially introducing a knowledge bias that narrows the model's generalization capabilities.
>
> - SlimPajama: To mitigate the limitations observed in the Wikitext-based adaptation, we also adapt the model by fine-tuning on the SlimPajama dataset, which comprises a mixture of texts from more diverse data sources like Commoncrawl, C4, GitHub, Books, ArXiv, Wikipedia, and StackExchange. Notably, using this dataset, the performance drop on MMLU is only 0.4% on average, with some categories (like 'Others') showing comparable or even slightly improved performance.
>
>
> |     | Dataset    | Humanities    | STEM    | Social Sciences    | Others    | Average    |
> |-------------|-------------|-------------|-------------|-------------|-------------|-------------|
> | LLaMa-2-7B | Unknown    | 42.9 | 36.4 | 51.2 | 52.2 | 45.3 |
> | MAGNET | Wikitext    | 41.1 | 33.5 | 49.5 | 52.1 | 43.7 |
> | MAGNET | SlimPajama    | 41.7 | 35.8 | 50.9 | 52.8 | 44.9 |
>
> So, all in all, we demonstrate that the model retains its text generation capabilities post-adaptation in the following ways:
> 1. We show that the model retains the knowledge and generation capabilities of the base model through experiments on the MMLU benchmark (as discussed above).
> 2. Compared to other text encoders and LLM-adaptation methods (like LLM2Vec), the models adapted using our approach are significantly less prone to repeating words and phrases  (Section 4.4).
> 3. Through human evaluation, we demonstrate that our method adapts LLM to perform text infilling, which is inherently a generative task (Section 4.3).
> 4. We also include qualitative examples in Table 10, showcasing the text generated from a seed prompt by our model and the baselines.
>
> We have included the MMLU experiment in the revised version of our paper and hope these findings substantiate our claim that the model largely retains its text-generation capabilities post-adaptation.
>
>
> > (Weakness 2) It's unclear whether the results reported in Section 4 reflect zero-shot performance or task-specific fine-tuning.
>
> **Ans.** All results, except for token-level classification (Table 1), are obtained in a zero-shot manner. For token-level tasks, we train a linear layer to enable the model to interpret class labels, which we believe is standard practice for evaluating such tasks.
>
>
> > (Question) In the "Overall Loss" section of Appendix A, it's unclear why a two-stage training approach is necessary and why only two objectives are used in the first stage.
>
> **Ans.** We begin with the MNTP and MSG tasks because they help the model incorporate future context, a capability that the base model lacks. This approach aligns with prior work [1], which adapts LLMs for representation learning.
>
> We regret not addressing this point in the original version of our paper and have now updated it accordingly. Thank you for bringing this to our attention!
>
> [1] BehnamGhader et al., LLM2Vec: Large Language Models Are Secretly Powerful Text Encoders, COLM 2024.

---

> ### Author Response · Authors · 2024-11-23
> **Request to Review the Rebuttal**
>
> Dear Reviewer V5cp,
>
> Thank you for providing valuable feedback on our submission. We have submitted our response to your concerns, along with a revised version of the paper. With only a few days remaining until the rebuttal deadline, we would greatly appreciate it if you could review our response and let us know if you have any additional questions or concerns. We are eager to ensure all your points are thoroughly addressed and look forward to your feedback.
>
> Best regards,
>
> Authors of Submission3348

---

> > ### Author Response · Authors · 2024-11-25
> > **Sincerely Awaiting Your Feedback**
> >
> > Dear Reviewer V5cp,
> >
> > As the discussion period concludes tomorrow, we kindly request you to review our response and share any remaining questions or concerns. We are committed to addressing all your points comprehensively and look forward to your valuable feedback.
> >
> > Best regards

---

> > > ### Author Response · Authors · 2024-11-27
> > > **Follow-Up Request for Feedback**
> > >
> > > Dear Reviewer V5cp,
> > >
> > > With the deadline to update the paper approaching in a few hours, we wanted to kindly follow up to see if our response addresses your concerns. If you have any remaining questions or suggestions, please let us know—we greatly value your feedback and are happy to address any additional points.
> > >
> > > Thank you for your time.
> > >
> > > Best regards

---

> ### Author Response · Authors · 2024-12-01
> **Follow-up on Reviewer Feedback**
>
> Dear Reviewer V5cp,
>
> We are following up on our previous responses addressing your concerns. We are eager to understand if our revisions meet your requirements or if you need any additional clarifications. Your specific feedback will help us finalize and improve our submission.
>
> We look forward to your guidance.

---

### Meta-Review · Area_Chair_8peq · 2024-12-20

**Metareview:**

The paper proposes to enhance generative decoders by balancing casual attention and bi-directional attention. Experiments were conducted on sequential labeling and text infilling tasks. Results show that the proposed approach is better than several baselines.

Reviewers generally give borderline or rejection scores. Several key drawbacks include lack of sufficient analysis on generation tasks and lack of comparison with previous work.

**Additional Comments On Reviewer Discussion:**

Reviewers generally give borderline or rejection scores. Even the reviewer who gave a borderline leaning positive acceptance pointed out the lack of generative experiments. This is a major concern.

---

### Decision · Program_Chairs · 2025-01-22

Reject